# Predictors of Testicular Cancer Mortality in Brazil: A 20-Year Ecological Study

**DOI:** 10.3390/cancers15164149

**Published:** 2023-08-17

**Authors:** Ana Paula de Souza Franco, Eric Renato Lima Figueiredo, Giovana Salomão Melo, Josiel de Souza e Souza, Nelson Veiga Gonçalves, Fabiana de Campos Gomes, João Simão de Melo Neto

**Affiliations:** 1Urogenital System Clinical and Experimental Research Unit, Institute of Health Sciences, Federal University of Pará (UFPA), Belém 66075-110, PA, Brazil; andyftkc@gmail.com (A.P.d.S.F.); eric.renatoo@gmail.com (E.R.L.F.); giovana.salomao@gmail.com (G.S.M.); josiel.souza1997@gmail.com (J.d.S.e.S.); 2Laboratory of Epidemiology and Geoprocessing of Amazon, State University of Pará (UEPA), Belém 66113-010, PA, Brazil; nelsonveiga@uepa.br; 3Ceres Medical School (FACERES), São José do Rio Preto 15090-305, PA, Brazil; facamposgomes@gmail.com

**Keywords:** testicular neoplasms, mortality registries, risk factors, epidemiology

## Abstract

**Simple Summary:**

The aim of this study is to investigate the factors that influence mortality and survival rates in testicular cancer in Brazil. The researchers examined sociodemographic and risk factors, as well as the influence of diagnostic and treatment procedures on reducing mortality. The study analyzed data from individuals who died of testicular cancer between 2001 and 2020. Results showed a progressive increase in mortality after 2011, particularly among individuals born after 1976 and aged 15 to 40 years. Certain regions, marital statuses, and racial affiliations were associated with higher mortality rates. The use of certain pesticides also affected mortality rates in different regions. Despite high rates of diagnostic procedures, they were not sufficient to reduce mortality. The study highlights the need for targeted interventions and improved treatment strategies to reduce testicular cancer mortality rates in Brazil.

**Abstract:**

Testicular cancer is common in young men, and early detection and multimodality treatment can lead to successful outcomes. This study aims to identify sociodemographic and risk factors associated with higher testicular cancer mortality and poorer survival rates, while examining the impact of diagnostic and treatment procedures on reducing mortality. The retrospective ecological study analyzed mortality data from testicular cancer in Brazil from 2001 to 2020. Sociodemographic variables such as marital status, age, birth period, year of death (cohort), race, and geographic region were assessed. Risk factors included cryptorchidism and pesticide exposure. Data were subjected to statistical analysis, which revealed an increasing trend in mortality after 2011 among persons born after 1976 in the 15–40 age group. Individuals in the South Region, whites, and singles had higher age-standardized mortality rates (ASMRs), while singles had lower survival rates. The Northeast region had a higher survival rate. Fungicides and insecticides increase ASMR in Brazil. Herbicides increase ASMR in the Northeast and Midwest regions and insecticides increase ASMR in the Northeast, Southeast, and Midwest regions. High rates of implementation of diagnostic procedures in the Midwest were not sufficient to reduce ASMR. No treatment procedure was associated with mortality at the national or regional level.

## 1. Introduction

Testicular cancer (TC) is the third leading cause of cancer death and the most common malignant neoplasm in young men aged 15–45 years, accounting for 1% of all male cancers and 5% of urological malignancies [1,2] and its incidence has been steadily increasing over the past three decades [3]. Testicular cancer encompasses a variety of different neoplasms, dependent on the cell of origin and the typical age of presentation [4]. Testicular tumors can be divided into those derived from germ cell neoplasia in situ (NCGIS) and those not derived from NCGIS [5]. Histologically, they are classified as seminoma, non-seminoma (yolk sac, embryonal carcinoma, choriocarcinoma, and teratoma), or mixed [5]. Among these types, germ cell tumors are the most common, represented by spermatocytic tumors, seminomas, which originate in the germinative epithelium of the seminiferous tubules, and non-seminomas [6]. Mortality from testicular cancer may be influenced not only by the stage of the disease, the aggressiveness of the histologic type of the tumor, resistance to standard chemotherapy with cisplatin, the occurrence of bilateral tumors, the occurrence of secondary malignancies, but also by the delay in establishing the correct diagnosis and initiating appropriate treatment [7,8].

Among the risk factors that are described in the literature for the development of testicular cancer and that may interfere with mortality in men is cryptorchidism, which is a congenital defect in which one or both testicles do not descend into the scrotum [9]. Studies show that boys with cryptorchidism have an overall relative risk of 4.8 (95% confidence interval 4.0–5.7) of developing testicular germ cell tumors [9]. However, even if the association between cryptorchidism and NCGIS is clinically established, the mechanisms that lead to carcinogenesis are still unknown [10]. It is possible to infer that there is a molecular genetic relationship between cryptorchidism and the development of NCGIS due to the occurrence of common genetic factors in the etiology of these two pathologies [6]. Another explanation is that the abnormal position of the testicle is itself responsible for germ cell tumorigenesis. Moreover, the interruption of a common regulatory pathway, such as androgen signaling, may be an implicit cause for the association of cryptorchidism and NCGIS [10]. In this sense, the results show that genetic mutations are relevant factors in testicular cancer and that several genes are associated with uncontrolled cell growth in this type of cancer, highlighting the KIT and KRAS genes, which are responsible for cell signaling; the CDC27 gene, which regulates the cell cycle; the XRCC2 gene, which is responsible for DNA repair; and p53, which is involved in the regulation of cell growth and programmed cell death [11].

Continued monitoring of these trends is essential for the development of appropriate prevention and treatment strategies worldwide. TC incidence and mortality are higher in countries with a medium to high Human Development Index, with a higher incidence in Northern Europe, especially Norway and Denmark, and increasing in many countries, especially in Southern Europe and Latin America, but decreasing in Northern Europe, the United States, and Australia [12]. Particularly in Brazil, studies show that the risk of cryptorchidism is increased in some states, such as São Paulo, Paraíba, Pernambuco, Sergipe, and Santa Catarina. Although the Southeast region has the highest prevalence, only the São Paulo federation units showed increased risk [13]. In addition, the use of pesticides is a factor that has been linked to this increased risk. Pesticides (for example, pesticides, fungicides, and herbicides) are present in the group of environmental endocrine disruptors (EDs) that comprise substances that mimic androgenic and estrogenic action, thus altering physiological hormone homeostasis [14].

Epidemiological studies have shown an association between maternal exposure to pesticides and adverse events in pregnancy. Previous studies have shown that postnatal exposure can have adverse effects on semen quality. These effects include decreased semen volume and percentage of sperm motility, increased semen pH, presence of morphological abnormalities in the sperm head, and increased leukocyte concentration [15]. In addition, pesticide exposure can affect hormone levels, resulting in decreased serum levels of luteinizing hormone and testosterone [16].

Although testicular cancers are limited to a small proportion of men, they are generally aggressive and have a strong psychological impact on patients. Therefore, early diagnosis is of fundamental importance, since early detection leads to reduced mortality, increased survival rates, and disease prognosis [17]. Currently, testicular and scrotum biopsy is one of the safest procedures for the detection of NCGIS. Although this procedure does not totally exclude the risk of cancer recurrence, false negatives (about 0.03%) and complications after biopsy are rare [18]. One can also cite as a diagnostic method the scrotal ultrasound which has high sensitivity to detect abnormalities. Once NCGIS affects the testicles, they present an irregular echo pattern, which may be related to the presence of hyaline bodies or testicular microliths. An ultrasound finding of an irregular echo pattern along with testicular microlithiasis should alert medical professionals to a malignancy. In addition, the ultrasonographic examination will identify small nonpalpable tumors [19].

Importantly, most primary and metastatic germ cell tumors secrete protein products that can be detected in the circulating blood. These serum biochemical tumor markers are very useful for diagnosis, staging, and risk assessment, as well as the evaluation of treatment response and relapse detection [20]. Among them is the alpha-fetoprotein (AFP), which is a single-chain glycoprotein, the element of the yolk sac responsible for tumor production. In a study of nearly 1500 testicular cancer patients, more than 60% of patients with non-seminoma testicular cancer manifested elevated AFP levels, making it the most commonly increased tumor marker in testicular cancer [21]. Serum tumor markers alone are not diagnostic of testicular cancer, but very high values in men rarely occur outside of testicular cancer.

Among other diagnostic methods, conventional computed tomography stands out as the standard in GCTs, besides guiding the treatment, because the management following primary chemotherapy depends on the size of the residual mass found in abdominal/pelvic computed tomography [21,22]. Finally, MRI can be considered as an alternative imaging method to conventional TC scans. A prospective study of 52 patients showed no inferiority of the MRI over TC scan with a reader sensitivity of 94% (95% CI 80 to 100) for the MRI and 98% (95% CI 87 to 100) for TC scan [21].

The primary treatment for TC is a unilateral or bilateral orchiectomy with ganglionic emptying for most cases where patients present with a testicular mass that is suspicious for malignancy on ultrasonography. Besides being the standard treatment, it also provides specimens for histological diagnosis [2]. In recent years, partial orchiectomy has emerged as a promising treatment alternative for testicular preservation in difficult clinical scenarios, such as cases in which ultrasound has revealed ambiguity. This approach is mainly considered for selected patients, including those with bilateral lesions, monorchid tumors, and individuals who are psychologically distressed or wish to preserve their fertility. In these cases, there are less-defined therapeutic options, making partial orchiectomy a viable alternative [23,24]. In addition, new treatments are being investigated, including immunotherapy and its combination, as well as hormonal therapy, which have shown promising results in previous studies [25,26]; however, they are not yet commonly recorded in public databases.

The cure rate of TC is the highest of any solid tumor and the improved survival rate is mainly due to effective chemotherapy. In most cases, treatment for localized cancer consists primarily of surgery, with the option for some patients to receive a short course of adjuvant chemotherapy to reduce the risk of recurrence. In addition, approximately 80% of patients with metastatic disease can be cured with systemic cisplatin-based chemotherapy. Unfortunately, a proportion of patients develop platinum resistance and experience disease recurrence [3].

There are currently a scarce number of studies focused on the mortality of men with testicular cancer in the geographic regions of Brazil. Considering the importance of testicular cancer for public health among men and the need for high-quality cancer services to improve survival and quality of life, a study on this topic is of the utmost importance. In view of this, this study aimed to analyze the sociodemographic and risk factors that predict higher rates of mortality from TC and interfere with survival rates. Furthermore, it was verified whether diagnostic and treatment procedures interfere in the reduction of mortality by TC.

## 2. Materials and Methods

### 2.1. Ethical Aspects

The study analyzed information from a public access secondary database on testicular cancer mortality. Since the information is in the public domain and can be freely accessed and used, obtaining ethical approval from a research ethics committee was not necessary as stipulated in the National Health Council guidelines, document number 510 (7 April 2016) [27].

### 2.2. Type of Study

This was an epidemiological, ecological, inferential, and descriptive study [28].

### 2.3. Population

This study analyzed secondary data of male patients who died from testicular cancer between 2001 and 2020, catalogued in the Brazilian Ministry of Health’s Health Information System.

### 2.4. Eligibility Criteria

Deaths that occurred between 2001 and 2020, classified in the ICD-10 category, with code C62 (malignant neoplasm of the testicles), were included. In addition, there is information on hospital and outpatient procedures published after 2008. Deaths that occurred outside the study interval and individuals in the “ignored” category in any of the studied variables were excluded from the analyses.

### 2.5. Database

Data collection for this study was conducted using the Open Data Portal of the Department of Informatics of the Unified Health System (DATASUS), which is a platform provided by the Brazilian Public Ministry. DATASUS served as an open-access secondary database, from which information on TC mortality, social and demographic variables, as well as outpatient and hospital procedures, was extracted. The data were obtained through various health information systems, including the Mortality Information System (MIS), the Hospital Information System (SIH), and the Outpatient Information System (SIA) [29].

Data collected by SIH and SIA from 2008 onward were considered for the analysis. In addition, data from the National Information System for Live Births (SINASC) on live births with cryptorchidism during the relevant period were included in the analysis using DATASUS. The study used the five political–administrative macro-regions of Brazil as the basis for the analysis [30]. Population data for each of these regions were retrieved from the database to calculate the crude specific mortality rate (CMR). The CMR was calculated by dividing the number of deaths in a given period by the total population. To account for differences in the age structure of the population, age at death was used as an adjustment variable to calculate the age-standardized mortality rate (ASMR). The ASMR was standardized using the standard population provided by the World Health Organization (WHO) and expressed per 100,000 population. Therefore, the number of deaths and the ASMR were considered exposure variables in the analysis. 

Data on pesticides were obtained through the pesticide marketing reports published annually since 2009 by the Brazilian Institute for the Environment and Renewable Natural Resources (IBAMA) website, of the Ministry of the Environment. Companies that sell pesticides and similar components registered in Brazil must submit annual reports on the quantity produced and sold, including imports and exports [31].

### 2.6. Outcomes

Variables of interest included the combination of age, period, and cohort, as well as ethnic–racial classification, marital status, geographic location, and diagnostic and treatment procedures. Demographic characteristics included various geographic regions of the country, including the North, Northeast, Southeast, South, and Midwest. Social variables included marital status (Single, Married, Widowed, or Legally separated), age, birth period, year of death (cohort), and racial classification based on skin color, with White, Black, Yellow, Brown, and Indigenous categories.

The diagnostic procedure variables were: scrotum biopsy, testicular biopsy, tomography of the pelvis and lower abdomen, scrotum ultrasound, magnetic resonance imaging of the pelvis and lower abdomen, and alpha-fetoprotein dosage. Regarding the treatment procedures, the following were studied: chemotherapy for testicular germ tumor (1st and 2nd line); uni- or bilateral orchiectomy with lymph node dissection; and partial resection of the scrotum.

The risk factors for TC analyzed were cryptorchidism and the use of pesticides (herbicides, insecticides, and fungicides). Cryptorchidism data were normalized by the number of male live births in the same period, both collected from the DATASUS database, while the incidence was calculated by multiplying the normalization by 10,000. Regarding the use of pesticides, it was necessary to normalize the data according to the size, in km^2^, of each region. This information was collected from the website of the Brazilian Institute of Geography and Statistics (IBGE) [29,30,31].

### 2.7. Statistical Analysis

The APC Web tool (Biostatistics Branch, National Cancer Institute, Bethesda, MD, USA) was employed to verify the influence of age, time of death, and birth cohort on deaths from HT. In this model, the following parameters were studied: net drift (annual percentage variability of age-adjusted expected rates over time); Long vs. Cross RR (cross-sectional age curve and longitudinal age curve indicating age effects); all rate ratios (RR; age-adjusted incidence pattern in each period [or cohort]); and local drifts representing cohort effects. In addition, Wald tests followed by chi-square (χ^2^) were performed to identify the statistically significant variables according to age, period, and cohort factors [32]. *p* values < 0.05 were considered statistically significant. The age range of the study population and the number of deaths were grouped in 5-year intervals, resulting in 17 age groups (from 0 to 4 years to 80 years and more), four periods (2001–2005 to 2016–2020), and 20 birth cohorts of 5 years each (1921–2016).

To analyze the comparative analysis, initially, the Shapiro–Wilk test was performed to verify the normality of the data. Subsequently, the descriptive analysis was expressed as measures of central tendency (median), dispersion (minimum, interquartile, and maximum deviation), and graphs (box plot). The Kruskal–Wallis non-parametric test was applied and the Dunnett’s test was applied as a post hoc test to evaluate the real hypothesis of statistically significant differences between the ASMR medians in relation to marital status, race, and geographic regions in multiple comparisons.

For survival analysis, data were collected from MIS to determine survival time, taking into account the interval between birth and death according to race, marital status, and macro-region. Kaplan–Meier curves were constructed for these variables, and the log-rank, Breslow, and Tarone–Ware tests were applied to compare the distributions of survival rates from the curves for the beginning, middle, and end groups. Variables that had a statistically significant *p*-value were included in the Cox proportional hazards univariate and multivariate (adjusted) model for survival time to test the relationship between patient survival time and predictors.

To verify the correlation between ASMR and the variables considered risk factors, and diagnostic and treatment procedure, the Pearson correlation coefficient (r) was calculated to analyze the parametric data and the Spearman correlation (rs) was used when the distribution was not parametric; therefore, the data do not show a normal distribution on the Gaussian curve. Pearson and Spearman results were classified as high (0.67–1.00), moderate (0.34–0.66), low (0.33–0.11), and insignificant (≤0.10).

## 3. Results

During the period from 2000 to 2019, 5.179 cases of mortality caused by TC were registered in the public health system in Brazil.

### 3.1. Age–Period–Cohort Effect

The APC analysis showed that age, period, and cohort factors have an influence on the TC mortality rate (Figure 1A). The graph of the relationship between the longitudinal and transversal curves represents the net drift, where an increase in the RR trend over time can be seen (Figure 1B). Regarding the effect of the period on mortality, there was an increasing trend after the year 2011 (Figure 1C). For the cohort effect, men who were born after 1976 showed an increase in estimated mortality (Figure 1D). Regarding the local drift, it was concluded that the estimated annual percentage change in the TC mortality rate increased in the age group between 15 and 40 years old (Figure 1E).

### 3.2. Sociodemographic Factors

Figure 2 shows the comparison of ASMR according to marital status, racial classification, and geographic region.

In Brazil, single individuals (0.71 [0.21–1.57]) had a higher ASMR than legally separated individuals (0.30 [0.60–3.63]) (Figure 2A). Regarding race, White individuals (0.6 [0.41–0.88]) were higher than Black (0.21 [0.05–1.50]), Yellow (0.00 [0.00–2.60]), and Indigenous (0.21 [0.00–0.73]). Furthermore, Yellow was inferior to Black and Brown (Figure 2B). The South region (0.97 [0.34–1.71]) had the highest ASMR rate compared to the others (Figure 2C).

### 3.3. Survival Analysis

Figure 3 shows the Kaplan–Meier curves according to racial classification (Figure 3A), marital status (Figure 3B), and geographic region (Figure 3C), being different in the three portions of the curve. Table 1 presents the Cox regression univariate and multivariate (adjusted) analysis for each category. Regarding marital status, single individuals had the lowest survival rate. Married or legally separated individuals had lower survival rates than widowed individuals. Regarding race, Indigenous people had lower survival rates than Whites, Blacks, and Yellows. Still, regarding geographic regions, the Northeast had a higher survival rate than the other regions.

### 3.4. ASMR and Risk Factors, Diagnosis, and Treatment

Table 2 shows the correlation between ASMR and risk factors, diagnosis, and treatment in TC. When analyzing the risk factors, the following positive correlations were found for ASMR: use of herbicides in the Northeast and Midwest; use of fungicides in Brazil; and use of insecticides in Brazil and in the Northeast, Southeast, and Midwest regions. Figure 4 illustrates the spatial distribution of these indicators. Regarding diagnostic procedures, the following positive correlations were found for ASMR: testicular biopsy in Brazil; tomography of the pelvis and lower abdomen with Northeast and Midwest region; “scrotum ultrasound” and “pelvic and lower abdomen magnetic resonance” with the Midwest; and alpha-fetoprotein levels with Brazil and the Midwest. No treatment procedure was associated with mortality at the national or geographic level.

## 4. Discussion

Testing for testicular cancer has become more common in recent years, which may have contributed to the decrease in mortality despite being the most recurrent malignant neoplasm in young men, and is among the most curable neoplasms when identified early and treated with a multimodal approach [33]. The regional discrepancies found in Brazil lead to different patterns in mortality distribution [34].

Studies that compare world rates show that Brazilian mortality rates are lower than those of some developed countries and some Latin American countries but with predictions of an increase until 2030 [34]. The ages between 15 and 40 years have the highest incidence, with an increase of 1.8% per year being observed, in addition to a continued rise in mortality after 2011 in individuals born after 1976. Unlike most cancers that occur in adulthood, the incidence of TC does not increase with age [35]. This involvement may be interconnected with the activity of sex hormones in this age group. The cases of death in ages over 50 may be related to the absence of clear and specific symptoms and signs in the early stages of the pathology, with the exception of a painless swelling or a unilateral nodule, making early detection difficult [36].

One of the causes associated with cancer mortality in a population is the occurrence and prevalence of risk factors related to the pathology [37]. For TC, ethnic characteristics are related to increased incidence, thus affecting mortality rates. A study that analyzed the SEER (surveillance, epidemiology, and end results) data revealed that Caucasians have the highest incidence rate (2.08:100,000) of TC [35]. Since the Southern region is predominantly of European descent, with a high proportion of Caucasians (more than 70% of the population), the phenotypic characteristics of the Southern Brazilian population could explain the high rates of mortality from HT in this region [35]. Another probable reason for the mortality trends in southern Brazil would be better-structured cancer epidemiologic surveillance services, resulting in better-quality death records [37].

As mentioned, race is a risk factor for the occurrence of TC. As in the literature, in this study, individuals considered white had a higher ASMR. Explanations that include differences in genetic factors, lifestyle or cultural factors, environmental factors, and variability in hormone exposures can be attributed to these results [38]. In this regard, a previous study in a European population highlights a four- to sixfold risk of TC in children of men with TC and siblings an eight- to tenfold risk. The higher rate in the sibling relationship versus parent–child may represent the complex genetic/environmental shared risk or an autosomal recessive or X-linked component of complex inheritance. Also, according to the study, TC-associated alleles are significantly more frequent in non-Hispanic white populations than in black populations [39], corroborating the incidence patterns of the disease, its genetic relationship, and the findings of this study.

Studies on dietary patterns and the incidence of testicular carcinoma have revealed a significant association between high total fat intake and an increased risk of developing mixed germ cell tumors. In this sense, the cultural diversity in the food structure of different ethnic–racial groups may be a factor for the observed differences in the incidence of TC. In the Western diet, fat intake is relatively high at 35 to 45% of total calories, whereas, in the Eastern diet, fat accounts for only about 20% of total calories [40]

Regarding marital status, being single is related to significantly worse overall and cancer-specific survival [41]. In contrast, marriage functions as a partnership that promotes mutual support and care, resulting in surveillance and better care pertinent to the partner’s health [42]. Evidence for the beneficial effect of being married on survival in TC patients is unknown, but the likely justification for these factors is a more favorable family environment, the willingness of patients to seek medical help, or other factors [43,44,45,46,47]. Unmarried patients have a significantly higher risk of having metastatic cancer, undertreatment, and death resulting from cancer [48,49]. In addition, unmarried patients with ASMR have a two- to threefold higher mortality rate when compared to married patients [47]. Although studies have shown worse survival indicators among widowers and unmarried men [50,51], our study found higher survival rates in this group. It is worth noting that our study is observational and ecological in nature, which means that it cannot prove that widowhood causes a higher survival rate for testicular cancer. However, it does suggest that there is an association between the two, and more research is needed to understand the reasons for this.

Regarding the geographic regions of Brazil, the population of the Northeast had a higher survival rate. According to a study that verified the standardized mortality rates for TC in Brazil and its geographic regions in the period from 2001 to 2015, the Northeast region showed a lower mortality rate [45], ratifying the results found in our study. This can be explained by relating a lower incidence in these individuals and, consequently, lower mortality. On the other hand, the trend analysis with projections until 2030 indicates that mortality should increase in this region [45], which serves as a warning, not only in this location but in several regions of the country, for greater investments in socioeconomic aspects and the general infrastructure of health services—improvements in access to primary prevention, early diagnosis, and effective treatments.

In the analysis of the influence of pesticides on the occurrence of TC, it was observed that exposure to fungicides and insecticides increased ASMR in Brazil. Brazil is one of the largest consumers of pesticides in the world, appearing on the international scene, along with the United States and China, as one of the main consumers of pesticides in the world [52]. According to the Brazilian Institute of the Environment, in 2020, the country surpassed the mark of 500 thousand tons of pesticides per year, and, from 2009 to 2020, there was an increase of ~217%. More specifically, sales of fungicides increased by 263%, and insecticides showed the highest sales volume, reaching a growth of 308.6% [31]. In general, Brazilian commodity producers, aiming to meet the demands with minimal losses, use a large number of pesticides [53].

The volume of pesticide sales in 11 Brazilian states in 1985 and the analysis of human reproductive disorders observed in the 1990s suggest that there are moderate to high correlations between infertility and hospitalization rates for various cancers, including testicular cancer [54]. Another study showed that pesticide sales in 1985 showed a weak to moderate correlation with testicular cancer mortality between the years 1996–1998, with cancer mortality rates being significantly higher in places with moderate to high pesticide sales [55]. Therefore, these studies corroborate our work as pesticide use is related to adverse health effects and its relationship with TC mortality.

As a more specific result of the relationship between the consumption of pesticides in each region and its consequence, it was observed that the use of herbicides increases ASMR in the Northeast and Midwest, while insecticides increase ASMR in the Northeast, Southeast, and Midwest regions. Herbicides are chemicals that reduce or eliminate weeds, i.e., plants that compete with the crop for water and nutrients. Currently, herbicides lead the list of the most commercialized active ingredients [56]. Among herbicides, glyphosate is the most commercialized component in Brazil (IBGE 2020), accounting for approximately 51% of all pesticides consumed in Brazil [33]. The relationship between the consumption of pesticides and the development of cancer and other diseases is already recognized by the WHO.

With regard to the Northeastern region, the development of agribusiness in Baixo Jaguaribe in Ceará led to the expansion of agribusiness and the alteration of the local landscape, substituting native vegetation for crops highly dependent on chemical inputs, including herbicides, thus generating problems due to the high use of these compounds in the region and their disposal into the environment [56]. We should also mention, as an example, the municipality of Limoeiro do Norte, which stands out as a major agribusiness hub due to high production rates, especially in irrigated fruit farming, with a high consumption of pesticides, including glyphosate herbicide, considered potentially carcinogenic by the International Agency for Research on Cancer (IARC) [56].

During the period of 2000–2010, a study was conducted to investigate the effects of pesticide exposure on the health of the population of the municipalities of Limoeiro do Norte, Quixeré, and Russas, located in the State of Ceará. The results of this study showed a statistically significant trend of increasing hospitalization rates for malignant diseases. The authors also observed that the rates of hospitalizations for neoplasms were 1.76 times higher in the study municipalities compared to the control area. The mortality rate for neoplasms was also higher in the study municipalities that had higher exposure to pesticides [56].

In the years 2012 to 2014, a survey was conducted on the use of pesticides in Brazil. Among the states with the highest consumption presented are Mato Grosso, Mato Grosso do Sul, Goiás, and São Paulo, which represented 44–92% more than the national average; these states are located within the Southeast and Midwest regions [54]. In 2019, the Southeast and Midwest regions achieved the highest concentration of agricultural commodity production, with rates of 30% for each region of total national production [57].

Pesticides have endocrine-disrupting properties; their induced toxic effects on the human reproductive system are directly related to the dose, route of exposure, frequency of exposure, and genotypic characteristics of the affected populations [58]. One study examined the association between occupational pesticide exposure and sperm quality in organophosphate pesticide sprayers. The results of this study showed that pesticide sprayers had significant reductions in age-adjusted semen volume, sperm motility, normal sperm morphology, serum luteinizing hormone levels, serum testosterone levels, and semen zinc concentration (a marker of prostate function), among other changes [59]. These findings provide additional evidence that occupational exposure to pesticides affects sperm quality and sex hormones, which may trigger a series of changes leading to the development of testicular neoplasia.

The sale of pesticides in different states does not mean their full use in the same region, especially in those with a more active economic profile, where they can be purchased for use elsewhere. This issue can be explained by knowing that economic trade links can lead pesticides to be purchased, for example, in the Southeast, to be used in certain areas of the Southern states. Thus, this regional exchange can interfere in some way in almost all the regions analyzed.

Thus, it was found that the high rate of diagnostic procedures in the Midwest was not enough to reduce ASMR. It is important to note that, in Brazil, oncology services are highly concentrated in large metropolitan areas with higher socioeconomic levels, mainly in the Southeast and South regions. Still, the Midwest region presents an intermediate condition [34]. This disproportion in resource distribution results in delayed diagnosis and treatment for patients residing in less privileged regions, leading to a worse prognosis when compared to those residing in large urban areas. In addition, the influx of patients migrating to larger cities puts additional pressure on referral medical centers. However, forecasts for 2030 indicate that decreasing mortality rates should occur for the Midwest region [35].

In examining the trends and projections of testicular cancer mortality in Brazil, different patterns were observed among the different geographic regions, revealing the epidemiologic diversity of the country. Understanding geographic and temporal variations in population levels and translating them into testicular cancer mortality rates is challenging without a clear identification of risk determinants. In this context, our ecological study provides an opening for the emergence of new hypotheses and research opportunities in future studies with different designs. This study has the following limitations: Because it extracts information from a secondary database, ecological bias may occur; omission of information from private services; and high racial miscegenation in the geographic regions of Brazil and racial classification defined by the team responsible for filling out the dying declaration, which may lead to divergent information. There is also the likelihood of delays or errors in data entry into the Ministry of Health system and the absence of variables such as immunotherapy treatments and their combinations, hormonal therapy, and clinical factors such as first-degree relatives, second-degree relatives, and previous history of cancer abnormalities. The databases used in our study do not provide this information, but we recommend that further studies investigate the impact of these variables. Finally, there is a lack of reports on pesticide use from the beginning of the period of this study (2001), which leaves a gap in the information.

## 5. Conclusions

There is a progressive increase in mortality after 2011, in individuals born after 1976, in the age group between 15 and 40 years old. In the South region, single and White individuals had a higher ASMR. Singles had the lowest survival rate. The Northeast region had a higher survival rate. Fungicides and insecticides increase ASMR in Brazil. The use of herbicides increases ASMR in the Northeast and Midwest, while insecticides increase ASMR in the Northeast, Southeast, and Midwest regions. The high rate of performance of diagnostic procedures in the Midwest was not enough to reduce ASMR. No treatment procedure was associated with mortality at the national or geographic level.

## Figures and Tables

**Figure 1 cancers-15-04149-f001:**
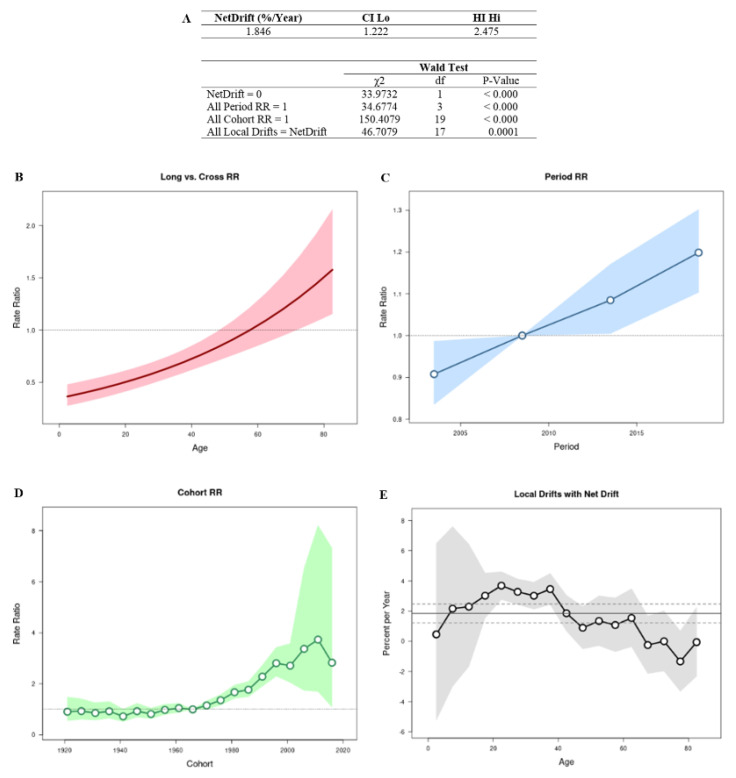
Age–period–cohort analysis using the Wald test (**A**), with analysis of mortality rates (**B**) by age, period RR (**C**), cohort RR (**D**), and local drift (**E**) analysis.

**Figure 2 cancers-15-04149-f002:**
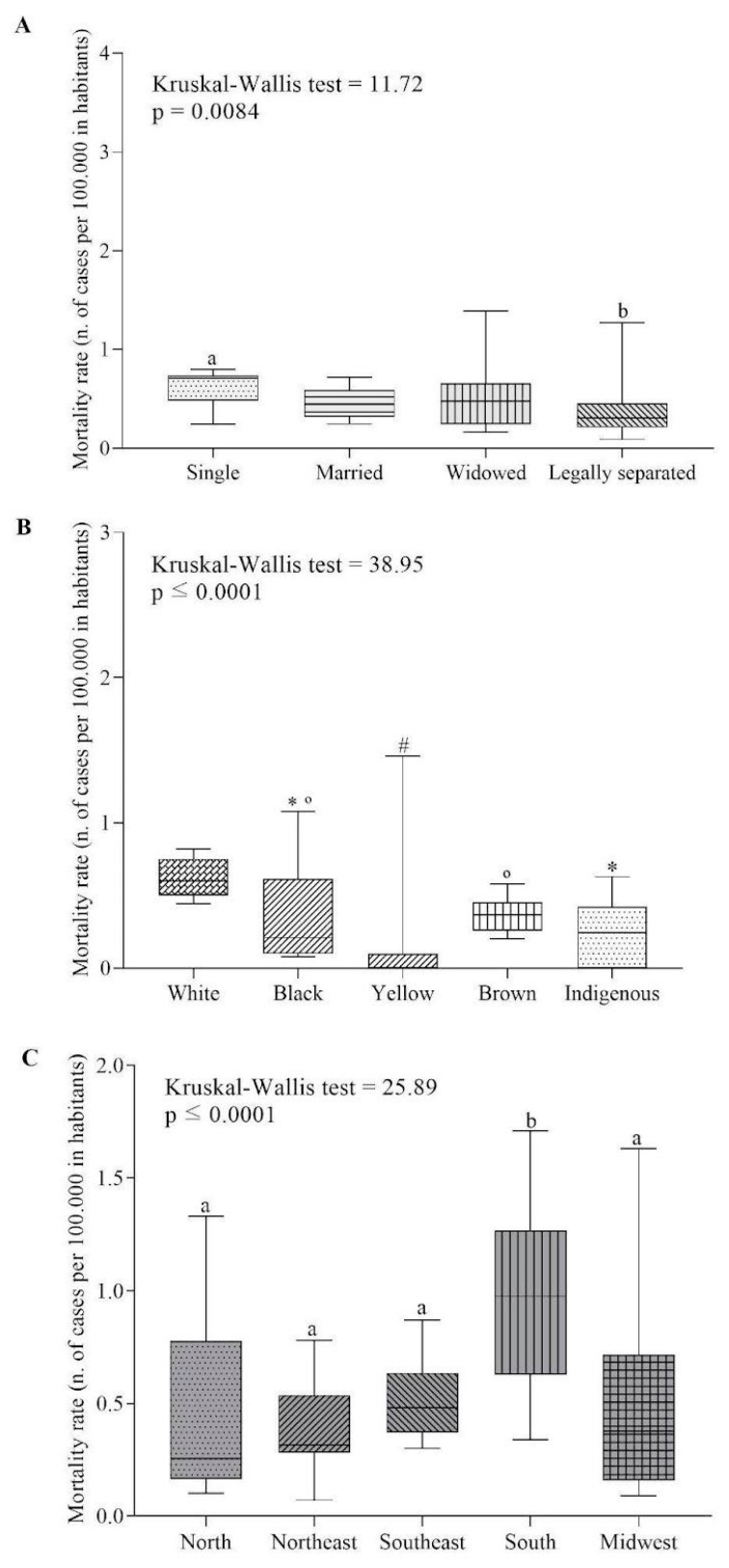
Testicular cancer mortality rate by marital status (**A**), race (**B**), and geographic region (**C**). ^a,b^ *p* < 0.05, post hoc Bonferroni. * *p* > 0.05, post hoc Bonferroni compared to White. ^#^ *p* < 0.05, post hoc Bonferroni compared to Black. ° *p* < 0.05, post hoc Bonferroni compared to Yellow.

**Figure 3 cancers-15-04149-f003:**
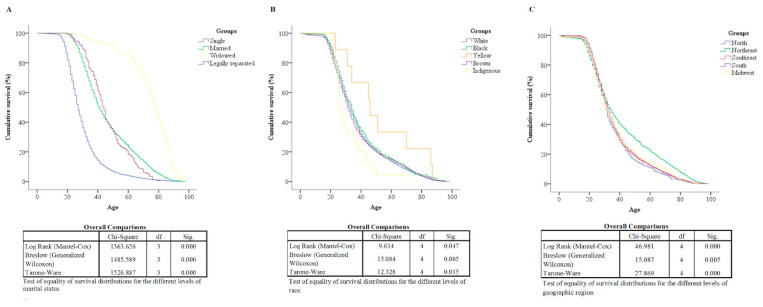
Survival analysis among different groups of marital status (**A**), racial classification (**B**), and geographic regions (**C**) using Kaplan–Meier.

**Figure 4 cancers-15-04149-f004:**
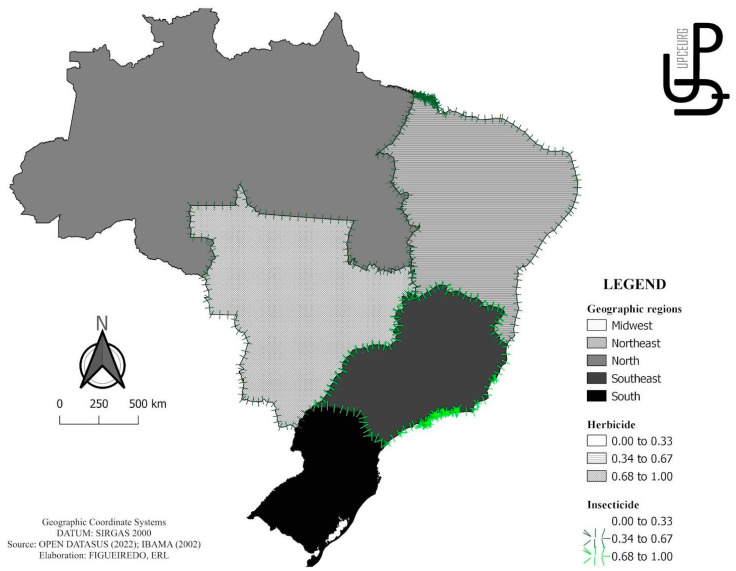
Significant (*p* < 0.05) positive correlations (r or rs) between the ASMR in geographic regions and the use of pesticides for testicular cancer.

**Table 1 cancers-15-04149-t001:** Cox’s proportional hazards univariate and multivariate (adjusted) models for survival time for individuals with testicular cancer according to marital status, race, and geographic regions.

		* p * ^ 1 ^	HR ^1^	CI ^1^	* p ^2^ *	** HR ^2^ **	** CI ^2^ **
Marital status							
Single (*n* = 924,153,300)	Married	0.000	0.391	0.367–0.416	0.474	1.022	0.962–1.087
	Widowed	0.000	0.150	0.126–0.178	0.226	1.113	0.936–1.322
	Legally separated	0.000	0.418	0.359–0.487	0.292	1.086	0.931–1.267
Legally separated (*n* = 43,026,684)	Married	0.386	0.934	0.801–1.090	0.480	0.941	0.805–1.111
	Widowed	0.000	0.358	0.286–0.448	0.836	1.024	0.818–1.282
Married (*n* = 560,319,997)	Widowed	0.000	0.328	0.285–0.391	0.861	0.971	0.999–1.349
Race							
White (*n* = 865,828,325)	Black	0.386	0.948	0.839–1.070	0.003	0.828	0.730–0.938
	Yellow	0.068	0.544	0.283–1.047	0.780	0.911	0.473–1.755
	Brown	0.466	1.023	0.962–1.088	0.000	0.793	0.737–0.853
	Indigenous	0.043	1.502	1.013–2.227	0.262	0.783	0.511–1.200
Indigenous (*n* = 8,208,013)	Black	0.027	0.631	0.419–0.950	0.805	1.057	0.680–1.642
	Yellow	0.009	0.362	0.169–0.766	0.704	1.163	0.533–2.536
	Brown	0.057	0.681	0.459–1.012	0.954	1.013	0.662–1.549
Brown (*n* = 824,445,607)	Black	0.276	0.932	0.821–1.058	0.564	1.077	0.836–1.388
	Yellow	0.073	0.549	0.285–1.057	0.720	1.299	0.311–5.426
Regions							
North (*n* = 161,365,743)	Northeast	0.000	0.716	0.633–0.810	0.635	0.969	0.851–1.104
	Southeast	0.126	0.918	0.822–1.024	0.071	1.117	0.991–1.261
	South	0.094	0.908	0.812–1.016	0.287	1.072	0.943–1.218
	Midwest	0.191	0.908	0.787–1.049	0.305	0.924	0.793–1.075
Midwest (*n* = 140,521,496)	Northeast	0.000	0.788	0.696–0.892	0.476	1.049	0.919–1.197
	Southeast	0.861	1.010	0.904–1.129	0.002	1.210	1.076–1.361
	South	0.998	1.000	0.893–1.120	0.017	1.161	1.027–1.312
Northeast (*n* = 518,192,220)	Southeast	0.000	1.281	1.179–1.391	0.568	0.909	0.654–1.263
	South	0.000	1.268	1.163–1.381	0.504	0.894	0.643–1.243
Southeast (*n* = 781,384,311)	South	0.757	1.010	0.946–1.079	0.517	0.897	0.644–1.248

^1^ Hazard ratio (HR) and CI: confidence interval of 95%. ^2^ adjusted hazard ratio (HR) and CI: confidence interval of 95%.

**Table 2 cancers-15-04149-t002:** Correlation between ASMR in different geographic regions and risk factors, diagnosis, and treatment for testicular cancer.

	Brazil	North	Northeast	Southeast	** South **	** Midwest **
Risk factors						
Cryptorchidism	*n* = 5980 r: 0.380 CI: −0.216–0.770 *p*: 0.200	*n* = 199 rs: −0.160 CI: −0.664–0.444 *p*: 0.599	*n* = 1606 r: 346 CI: −0.253–0.753 *p*: 0.246	*n* = 3030 r: 0.076 CI: −0.496–0.602 *p*: 0.805	*n* = 845 r: −0.139 CI: −0.641–0.446 *p*: 0.651	*n* = 300 r: 0.089 CI: −0.486–0.610 *p*: 0.773
Pesticides						
* Herbicide *	FPU = 0.884 rs: 0.340 CI: −0.308–0.773 *p*: 0.277	FPU = 0.465 r: −0.314 CI: −0.752–0.317 *p*: 0.321	FPU = 2.000 r: 0.670 CI: 0.157–0.899 *p*: 0.017	FPU = 3.044 r: 0.384 CI: −0.243–0.785 *p*: 0.217	FPU = 4.535 r: −0.040 CI: −0.600–0.546 *p*: 0.902	FPU = 3.012 r: 0.706 CI: 0.221–0.911 *p*: 0.010
* Fungicide *	FPU = 0.110 rs: 0.730 CI: 0.250–0.922 *p*: 0.009	FPU = 0.052 r: −0.323 CI: −0.757–0.308 *p*: 0.305	FPU = 0.211 rs: 0.210 CI: −0.430–0.709 *p*: 0.509	FPU = 0.988 rs: −0.175 CI: −0.691–0.459 *p*: 0.588	FPU = 1.013 r: 0.098 CI: −0.505–0.635 *p*: 0.765	FPU = 0.719 r: 0.401 CI: −0.224–0.79 *p*: 0.196
* Insecticide *	FPU = 0.065 r: 0.755 CI: 0.319–0.927 *p*: 0.004	FPU = 0.041 r: −0.319 CI: −0.755–0.312 *p*: 0.312	FPU = 0.223 r: 0.637 CI: 0.100–0.887 *p*: 0.026	FPU = 0.503 r: 0.715 CI: 0.239–0.914 *p*: 0.009	FPU = 0.509 r: −0.080 CI: −0.625–0.517 *p*: 0.804	FPU = 0.569 r: 0.600 CI: 0.040–0.873 *p*: 0.039
Diagnostic						
Scrotal biopsy	*n* = 2270 rs: −0.164 CI: −0.666–0.441 *p*: 0.590	*n* = 26 rs: −0.510 CI: −0.834–0.075 *p*: 0.077	*n* = 117 rs: 0.061 CI: −0.521–0.604 *p*: 0.842	*n* = 1948 rs: −0.105 CI: −0.631–0.487 *p*: 0.735	*n* = 163 r: 0.231 CI: −0.367–0.693 *p*: 0.448	*n* = 16 rs: −0.018 CI: −0.576–0.551 *p*: 0.955
Testicular biopsy	*n* = 144 rs: 0.583 CI: 0.028–0.863 *p*: 0.040	*n* = 10 rs: −0.120 CI: −0.640–0.476 *p*: 0.694	*n* = 9 rs: −0.108 CI: −0.633–0.485 *p*: 0.724	*n* = 74 rs: 0.355 CI: −0.260–0.766 *p*: 0.231	*n* = 34 rs: 0.050 CI: −0.528–0.597 *p*: 0.870	*n* = 17 rs: 0.226 CI: −0.387–0.701 *p*: 0.456
Tomography of the pelvis and lower abdomen	*n* = 3.198.010 r: 0.452 CI: −0.132–0.803 *p*: 0.121	*n* = 177.563 r: −0.068 CI: −0.597–0.502 *p*: 0.826	*n* = 445.153 r: 0.582 CI: 0.045–0.858 *p*: 0.037	*n* = 1.756.486 r: 0.185 CI: −0.407–0.668 *p*: 0.544	*n* = 581.534 r: 0.001 CI: −0.550–0.5522 *p*: 0.997	*n* = 237.274 r: 0.598 CI: 0.071–0.864 *p*: 0.031
Scrotal ultrasound	*n* = 982.335 r: 0.552 CI: −0.017–0.851 *p*: 0.053	*n* = 77.269 rs: 0.033 CI: −0.540–0.586 *p*: 0.916	*n* = 189.563 rs: 0.459 CI: −0.141–0.812 *p*: 0.115	*n* = 503.243 rs: 0.293 CI: −0.324–0.735 *p*: 0.331	*n* = 157.771 rs: 0.147 CI: −0.455–0.656 *p*: 0.630	*n* = 54.489 rs: 0.592 CI: 0.043–0.867 *p*: 0.036
Magnetic resonance imaging of the pelvis and lower abdomen	*n* = 293.066 r: 0.486 CI: −0.088–0.818 *p*: 0.092	*n* = 22.723 r: −0.069 CI: −0.597–0.501 *p*: 0.822	*n* = 51.823 r: 0.494 CI: −0.079–0.821 *p*: 0.086	*n* = 149.297 r: 0.234 CI: −0.364–0.695 *p*: 0.442	*n* = 48.868 rs: −0.005 CI: −0.567–0.560 *p*: 0.989	*n* = 20.355 r: 0.625 CI: 0.112–0.875 *p*: 0.022
Alpha-fetoprotein Dosage	*n* = 51.238 rs: 0.631 CI: 0.105–0.881 *p*: 0.024	*n* = 706 rs: −0.215 CI: −0.695–0.396 *p*: 0.476	*n* = 1.285 rs: 0.435 CI: −0.170–0.802 *p*: 0.137	*n* = 29.425 rs: 0.326 CI: −0.291–0.752 *p*: 0.273	*n* = 19.638 rs: 0.081 CI: −0.506–0.616 *p*: 0.790	*n* = 184 rs: 0.564 CI: 0.001–0.856 *p*: 0.047
Treatment						
First-line chemotherapy for testicular germ tumor	*n* = 39.298 r: 0.286 CI: −0.315–0.723 *p*: 0.344	*n* = 1.588 r: 0.309 CI: −0.291–0.735 *p*: 0.303	*n* = 4.178 r: 0.424 CI: −0.165–0.791 *p*: 0.148	*n* = 18.033 r: 0.037 CI: −0.525–0.576 *p*: 0.904	*n* = 13.083 r: −0.057 CI: −0.589–0.510 *p*: 0.853	*n* = 2.416 r: 0.312 CI: −0.289–0.736 *p*: 0.300
Second-line chemotherapy for testicular germ tumor	*n* = 9.938 r: 0.126 CI: −0.456–0.633 *p*: 0.680	*n* = 316 r: 0.532 CI: −0.027–0.837 *p*: 0.061	*n* = 1.025 r: 0.290 CI: −0.310–0.725 *p*: 0.336	*n* = 4.145 r: 0.220 CI: −0.377–0.687 *p*: 0.470	*n* = 3.829 r: −0.313 CI: −0.737–0.287 *p*: 0.298	*n* = 623 r: −0.101 CI: −0.617–0.477 *p*: 0.743
Uni- or bilateral orchiectomy with lymph node dissection	*n* = 1989 r: −0.495 CI: −0.822–0.077 *p*: 0.086	*n* = 107 rs: 0.300 CI: −0.317–0.739 *p*: 0.317	*n* = 401 rs: −0.520 CI: −0.838–0.061 *p*: 0.070	*n* = 980 r: −0.214 CI: −0.684–0.382 *p*: 0.483	*n* = 370 r: −0.235 CI: −0.696–0.363 *p*: 0.439	*n* = 131 r: −0.308 CI: −0.734–0.293 *p*: 0.306
Partial resection of the scrotum	*n* = 1685 r: 0.226 CI: −0.372–0.691 *p*: 0.459	*n* = 136 r: 0.374 CI: −0.222–0.767 *p*: 0.207	*n* = 668 r: 0.457 CI: −0.126–0.805 *p*: 0.117	*n* = 511 r: 0.143 CI: −0.443–0.643 *p*: 0.642	*n* = 286 r: −0.040 CI: −0.5785–0.522 *p*: 0.896	*n* = 84 rs: 0.332 CI: −0.284–0.755 *p*: 0.265

CI: Confidence interval of 95%. Pearson (r) or Spearman (rs) correlation coefficient. FPU: fee per use.

## Data Availability

All data were made available by https://opendatasus.saude.gov.br/dataset and https://www.gov.br/ibama/pt-br/assuntos/quimicos-e-biologicos (accessed on 12 June 2023).

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
