# Peer review of "Predictors of Testicular Cancer Mortality in Brazil: A 20-Year Ecological Study"

_cancers, 2023, doi:10.3390/cancers15164149_

Round 1
Reviewer 1 Report
This is an interesting study related to mortality for testis cancer in Brazil. The paper is well written and easy to follow. Data presentation is OK and the reference list is updated.
Minor corrections are requered:
Abstract
Line 23 …rates, hile examining the im….please correct spelling using while instead of hile...
These two sentencies look contradictory to me. I would suggest to review them and clarify....
Line 30, abstract, ...Survival rates were lower in the Northeast region
Line 456, conclussion,... The Northeast region had a higher survival rate.
Author Response
Point: This is an interesting study related to mortality for testis cancer in Brazil. The paper is well written and easy to follow. Data presentation is OK and the reference list is updated.
Minor corrections are requered:
Abstract
Line 23 …rates, hile examining the im….please correct spelling using while instead of hile...
These two sentencies look contradictory to me. I would suggest to review them and clarify....
Line 30, abstract, ...Survival rates were lower in the Northeast region
Line 456, conclussion,... The Northeast region had a higher survival rate.
Response: We would like to thank the reviewer for the careful evaluation of our manuscript on testicular cancer mortality in Brazil. We are pleased to know that the study is considered interesting and that the paper is well-written and easy to follow. We also appreciate the recognition of the updated reference list and the satisfactory data presentation.
Regarding the minor corrections, we have carefully considered and addressed each suggestion. We have made the necessary changes to the Abstract, correcting the spelling of "while" in line 23. In addition, we have carefully examined the sentences in question to ensure clarity and coherence in context. As a result, we have revised and clarified the statements accordingly.
Once again, we would like to thank the reviewer for his valuable feedback, which has undoubtedly helped to improve our work. We are confident that the revised version is now a more accurate and coherent representation of our findings. We hope that the revised manuscript meets the standards for publication, and we welcome any further feedback or suggestions for refinement, if necessary.

Reviewer 2 Report
The article entitled 'Predictors of testicular cancer mortality in Brazil: a 20-year eco- 2 logical study' is interesting, but the authors could improve the article.
1. The authors can discuss about the test prevalence during the time periods indicated as part of the study. Any history of testing that was performed in individuals or it could attributed to early diagnosis due to rapid testing at later years.
2. The authors could comment on the signaling pathways that are attributed to the testicular cancer.
3. What is the worldwide occurence of this cancer, the authors could include some statistics about it apart from Brazil data.
4. In Figure 3A, it looks like it is prevalent in widowed persons, what could be the contributing factor for this.
none
Author Response
Point:
The article entitled 'Predictors of testicular cancer mortality in Brazil: a 20-year eco- 2 logical study' is interesting, but the authors could improve the article.
1. The authors can discuss about the test prevalence during the time periods indicated as part of the study. Any history of testing that was performed in individuals or it could attributed to early diagnosis due to rapid testing at later years.
2. The authors could comment on the signaling pathways that are attributed to the testicular cancer.
3. What is the worldwide occurence of this cancer, the authors could include some statistics about it apart from Brazil data.
4. In Figure 3A, it looks like it is prevalent in widowed persons, what could be the contributing factor for this.
Response: We agree with the reviewer's suggestions. We have improved the article as follows and ask you to note these and the recommendations of the other reviewers in the text of the highlighted manuscript:
1. Thank you for your comment. We discuss the prevalence of testing during the time periods reported in the study. In particular, we note that testing for testicular cancer has become more common in recent years. See in Discussion.
2. Thank you for your comment. We comment on the pathways attributed to testicular cancer.
In particular, we note that testicular cancer is often associated with mutations in the KIT, KRAS, CDC27, XRCC2, and p53 genes. See in Introduction.
3. Thank you for your comment. We provide some statistics on the worldwide geographic incidence of testicular cancer. See in Introduction.
4. Thank you for your comment. We discuss the possible reason for the higher prevalence in widowed men. We note that there is no evidence to support this finding, but point out that this is due to the limitations of an ecological study and that studies with different designs are needed to test this hypothesis. Check in Discussion.

Reviewer 3 Report
Very interesting article.
An interesting data in this submitted article is the significant increase in testicular
cancer mortality during the last 10 years in Brazil. This is probably a specific figure
for a certain region of this country.
The mortality of testicular tumors can be influenced by the stage of the disease, the
aggressiveness of the histological type of the tumor, resistance to standard
chemotherapy with cisplatin, the occurrence of bilateral tumors, the occurrence of
secondary malignancies, but also the delay in establishing the correct diagnosis and
starting adequate treatment.
I recommend considering the use of the following literary sources that support
these claims.
1. Ondrušová, M., Ondruš, D.: Epidemiology and treatment delay in testicular
cancer patients: a retrospective study. International Urology and Nephrology,
2008, 40 (1): 143-148.
doi: 10.1007/s11255-007-9245-3
2. Mriňáková, B., Trebatický, B., Kajo, K. et al.: Bilateral testicular germ cell
tumors - 50 years experience. Bratislava Medical Journal. 2021; 122 (7): 449-53.
doi: 10.4149/BLL_2021_074
Author Response
Point:
Very interesting article.
An interesting data in this submitted article is the significant increase in testicular
cancer mortality during the last 10 years in Brazil. This is probably a specific figure
for a certain region of this country.
The mortality of testicular tumors can be influenced by the stage of the disease, the
aggressiveness of the histological type of the tumor, resistance to standard
chemotherapy with cisplatin, the occurrence of bilateral tumors, the occurrence of
secondary malignancies, but also the delay in establishing the correct diagnosis and
starting adequate treatment.
I recommend considering the use of the following literary sources that support
these claims.
- Ondrušová, M., Ondruš, D.: Epidemiology and treatment delay in testicular
cancer patients: a retrospective study. International Urology and Nephrology,
2008, 40 (1): 143-148.
doi: 10.1007/s11255-007-9245-3
- 2. Mriňáková, B., Trebatický, B., Kajo, K. et al.: Bilateral testicular germ cell
tumors - 50 years experience. Bratislava Medical Journal. 2021; 122 (7): 449-53.
doi: 10.4149/BLL_2021_074
Response: Thank you for your comment. We have improved our article with the information and reference you provided. Another demand has been met as per your and other reviewers' requests. Please check the highlighted text in the attatchment.

Reviewer 4 Report
The article written by Ana Paula de Souza Franco et al. describes “Predictors of testicular cancer mortality in Brazil: a 20-year ecological study” Overall, this study was well executed, and the data are attractive and exciting to the readers of this journal. I recommend this manuscript for publication after addressing the following comments.
1. Novel treatment for TC, such as immunotherapy and its combination, and hormone therapy should be included. These treatments have demonstrated satisfactory positive results and may significantly impact future predictions.
2. The fonts in Figure 1 appear to be small. Can you make it clear? Other figures are also in low resolution.
3. Can you explain why the relation to Case (e.g., first-degree relative, second-degree relative), prior history of TC abnormality (e.g., orchitis) are not included in the Cox’s proportional hazards univariate model since they are critical factors?
Please correct the grammatical and spelling errors in the manuscript.
Author Response
Point 1: Novel treatment for TC, such as immunotherapy and its combination, and hormone therapy should be included. These treatments have demonstrated satisfactory positive results and may significantly impact future predictions.
Response: Thank you for your comment. Indeed, novel treatments for TC, such as immunotherapy and its combination and hormonal therapy, have shown positive results and may have a significant impact on future predictions, We have included this information in our Introduction to improve the context and thematic presentation of our object of study according to their valuable contributions. Please read the highlighted text in the Introduction.
However, the absence of a new treatment variable in our study is due to a limitation of the health system in which the data were collected. The health system in question does not provide data on these new treatments.
Despite this limitation, we believe that our results are still meaningful and can be used to develop more effective prevention and treatment strategies for this condition.
We mention this limitation of our study more in the Discussion section, please check the highlighted text. We welcome your suggestions for improvement and hope that you will consider our paper for publication.
Point 2: The fonts in Figure 1 appear to be small. Can you make it clear? Other figures are also in low resolution.
Response: Thank you for your comment. We confirm that this figure, as well as the entire study, meets the journal's mandatory resolution standards of 300 dpi. We believe that the image in question has lost resolution in the document file.
Point 3: Can you explain why the relation to Case (e.g., first-degree relative, second-degree relative), and prior history of TC abnormality (e.g., orchitis) are not included in the Cox’s proportional hazards univariate model since they are critical factors?
Response: Thank you for your comment.
You are correct in stating that the relationship to the case (e.g., first-degree relative, second-degree relative) and history of CT abnormality are critical factors that may affect the survival of CT patients. However, these factors were not included in the univariate Cox proportional hazards model because they were not available in the data from the health system where we collected our data. The health system in question does not provide data on these variables.
We will mention this limitation of our study in the Discussion section. We welcome your suggestions for improvement and hope that you will consider our article for publication.
